# Relationship Between GPS-Derived Variables and Subjective Questionnaires Among Elite Youth Soccer Players

**DOI:** 10.3390/sports13080246

**Published:** 2025-07-25

**Authors:** Krisztián Havanecz, Péter János Tóth, Bence Kopper, Csaba Bartha, Sándor Sáfár, Marcell Fridvalszki, Gábor Géczi

**Affiliations:** 1Training Theory and Methodology Research Center, Hungarian University of Sports Science, 1123 Budapest, Hungary; bartha.csaba@tf.hu (C.B.); safar.sandor@tf.hu (S.S.); 2Department of Sport Games, Hungarian University of Sports Science, 1123 Budapest, Hungary; toth.peter.janos96@gmail.com; 3Faculty of Kinesiology, Hungarian University of Sports Science, 1123 Budapest, Hungary; kopper.bence@tf.hu (B.K.); fridvalszki.marcell@tf.hu (M.F.); 4Department of Sport Management, Hungarian University of Sports Science, 1123 Budapest, Hungary; geczi.gabor@tf.hu

**Keywords:** GPS, perceived wellness, RPE, training load, youth soccer

## Abstract

The aim of this study was to examine the relationship between the external load (EL) and internal load among U15, U17, and U19 youth soccer players and to identify the factors best influencing the rating of perceived exertion (RPE) and session-RPE (s-RPE) from Global Positioning System-derived variables. Data were collected from 50 male youth soccer players over an 11-week in-season period, encompassing a total of 1386 observations (145 training sessions and 33 matches). The findings indicate that during training sessions, the relationship between EL-derived volume variables and s-RPE exhibited moderate-to-very-strong correlations (U15—r ranging from 0.23 to 0.52; U17—r ranging from 0.51 to 0.78; U19—r ranging from 0.34 to 0.61, *p* < 0.001). The strongest relationships were observed with the total distance, acceleration, deceleration, and player load variables (*p* < 0.001). However, perceived wellness measures showed weak correlations with almost every EL parameter. Considering matches for all age groups, total distance showed moderate-to-large correlation with s-RPE (ranging from 0.41 to 0.59, *p* < 0.001). Additionally, RPE and s-RPE were significantly influenced by the variables of total distance, acceleration, deceleration, medium-speed running per minute, sprint distance per minute, and deceleration per minute.

## 1. Introduction

The effects of training load (TL) on the human body can be divided into physiological and psychological responses. Physiological load is further categorized into external load (EL) and internal load (IL) [1]. EL is defined as the work completed by the athlete [2], while IL represents the relative biological stresses imposed on players during training sessions and matches [3]. A better understanding of TL may benefit practitioners in developing soccer players [4,5]. Also, given that young footballers undergo significant natural growth and maturation, TL monitoring differs from that of adults [6] and should not be defined by the standards applicable to fully grown athletes [7] as young athletes are not merely “small adults” [8].

Still, one of the most challenging aspects of determining TL is to accurately monitor players’ stress levels on the field, whether considering EL or IL and their convergence. With the advent of the Global Positioning System (GPS), data on EL-based physical demands of elite youth footballers can be monitored in relation to their matches and training sessions [9,10,11]. These measurements provide better insights when combined with systems known as the Global Navigation Satellite System (GNSS), which accurately calculates the player’s position on the field [12] at a higher sampling frequency of 10 Hz [13]. The integration of a triaxial accelerometer, magnetometer, and gyroscope enables users to see the mechanical TL (in the X-Y-Z axes) next to locomotor TL. Previously, numerous GPS-based variables were suggested to identify EL, such as total distance, high-speed running, sprint distance, acceleration, deceleration, and player load [14,15,16,17].

In relation to TL, the use of subjective questionnaires [18,19] offers a comprehensive insight into IL. For this, rating of perceived exertion (RPE) was developed by Borg [20] and later adapted by Foster et al. [21]. Additionally, another method (session-RPE) to quantify IL is to multiply the total duration (in minutes) by the RPE [21,22]. Marynowicz et al. [4] referred to it as a useful, cost-effective TL measurement in youth soccer. RPE is a valid and reliable tool to assess training intensity in youth soccer [23,24]. Interestingly, the s-RPE method is better to use for young athletes compared to adults [25], which indicates its utility. However, it should be noted that RPE values only provide feedback on perceived exertion and are not a substitute for objective IL measurements, such as heart rate measurements. Despite its advantage, there are concerns about its accuracy that must be considered due to complex interactions [26], such as hormone and substrate levels, psychological state, previous experience, and the environment [27,28,29], as well as subjectivity and the influence of the player’s mood. Additionally, the wellness questionnaire provides feedback on the state of wellness, which was proposed by Hooper and Mackinnon [30]. This contains question items such as sleep quality, fatigue, muscle soreness, mood, and stress level [31].

From a coaching perspective, it is essential to understand the relationship between EL and IL [32]. In the long term, it may help to plan their training and TL by implementing monitoring strategies in everyday practice. Recently, the combined use of GPS and subjective exertion measures has garnered increased attention in both young [4,33,34] and adult populations [35,36]. Previous research found that s-RPE and EL-based variables correlate well [37] but with a higher intensity, variables show weaker relationships [38], which suggest that there may be a difference in the relationship between TL during training and matches. In addition, moderate-to-very-large correlations were observed among elite youth soccer players [34], indicating that there could be a significant difference between EL and IL in older youth age groups. They also found a small correlation between perceived wellness responses and EL measures among elite youth soccer players. It has been previously stated that weak relationships exist between the total distance, accelerations, decelerations, and sleep quality and fatigue [39]. In addition, previous research has found a small-to-moderate relationship between intermittent small-sided games, and a moderate correlation between stress and the total distance among amateur soccer players [40].

However, previous research has typically focused on a single age group within a football academy over a one-season follow-up period. It is crucial to examine multiple age groups as differences in TL may be detected [19].

Thus, the aims of this study are (i) to explore the differences of the relationships between EL-derived variables, including total distance, medium-speed running distance, high-speed running distance, sprint distance, distance in accelerations, distance in decelerations, number of inertial movements, the player load in arbitrary units (AU), and their respective values expressed as per minute and IL variables, such as RPE, s-RPE, and perceived wellness across U15, U17, and U19 age groups; and (ii) to identify the GPS-derived variables that best influence RPE and s-RPE.

## 2. Materials and Methods

### 2.1. Experimental Design

A total of 1386 observations were made (U15—training sessions 357, matches 60; U17—training sessions 388, matches 62; U19—training sessions 446, matches 73), including 145 soccer training sessions (U15—49, U17—46, and U19—50) and 11 matches per team (Table 1). Additionally, 19 training sessions were excluded from the analysis out of 164 training sessions due to some indoor activities as the data source was deemed unreliable due to inadequate satellite connection. This study lasted from February to May during the 2023–2024 competitive period. The data accumulation for this study lasted 11 weeks. On average, players trained 4 to 5 times a week.

Training sessions occurred on weekdays between 15:30 and 17:00 and were performed on artificial pitches. They consisted of technical, tactical, and conditional elements (in the latter case, compensation running bouts). Additionally, there was a strength and conditioning program on Monday, Tuesday, and Thursday for all teams, which took 30 min before ball training. The warm-up (FIFA 11+ protocol) was taken into account in the data analysis, next to the ball training, although the cool-down was not. The microcycle structure (one microcycle meant one week) was considered as the number of days before or after a match day (MD±) [41]. On MD-6/MD+1 (6 days before Match Day/1 day after Match Day), there was a day off for every team. MD-5/MD+2 was considered a recovery session with tactical development. MD-4/MD+3 was primarily focused on small-sided games with 2v2, 3v3, 4v4, and 5v5 to develop speed, agility, or power. MD-3 aimed to develop strength–endurance with large-sided games. MD-2 was focused on technical and further tactical development. A tapering method (decreasing TL) was also used on this day, although not on a regular basis. MD-1 was an activation day (neuromuscular) with less-volume TL.

Regular participation was required in both training sessions and matches. This study included players who participated in training sessions 80% of the time and all activities were completed. We determined a minimum of 70 min of playing time in matches for the U15 age group, while at least 80 min participation was required for the U17 and U19 age groups [42].

### 2.2. Participants

A total of 50 male academy soccer players (N_all_ = 50) participated in this study, comprising U15 (N_U15_ = 18, mean ± SD: age 14.77 ± 0.25 years, height 173.21 ± 6.11 cm, body mass 59.09 ± 4.95 kg), U17 (N_U17_ = 16, mean ± SD: age 16.70 ± 0.27 years, height 176.60 ± 6.36 cm, body mass 65.02 ± 5.35 kg), and U19 (N_U19_ = 16, mean ± SD: age 17.69 ± 0.53 years, height 180.53 ± 6.47 cm, body mass 71.99 ± 6.45 kg). Goalkeepers were excluded from this study due to a lack of GPS sensors. Also, there were players from the U15 age group who played a couple of times in the U16 team so their data were also excluded. Another necessary exclusion criterion was that only participants with no known injuries during the study were included. To control for confounding variables, such as dehydration, players were allowed to drink still water during training sessions’ resting periods. All players were familiarized with the monitoring procedures used during this study, which are regular, everyday tasks for them. This research received institutional ethical approval where it was conducted. This study conformed to the stipulations of the Declaration of Helsinki (1964). In the case of academy football players, no informed consent was required as the data used in this study were collected as part of their daily routine monitoring [43].

### 2.3. External Load Measures

During both training sessions and matches, a portable GPS sensor was used to monitor each player’s physical activity (Catapult S7 Vector, Catapult Sports Ltd., Melbourne, Australia). The validity and reliability of these units were previously reported as having a typical error of measurement of 1.3% [13]. All players wore the same type of GPS device to avoid inter-unit error [44]. Furthermore, regarding the accuracy of the collected data, the average horizontal dilution of precision (HDOP) was 0.72 ± 0.1, an average of 15 ± 0.8 satellites were connected during data collection, and the average GNSS quality was 70% ± 1.9%, which is in line with the previous literature [45]. Otherwise, the datasets were excluded. This device provides locomotor load variables at a sampling rate of 10 Hz. It also includes integrated micro inertial sensors, such as a triaxial accelerometer, gyroscopes, and magnetometer, providing mechanical load variables at a sampling rate of 100 Hz. The chosen variables for this research were the following (both locomotor and mechanical variables are presented, units of measurements: meters, number): total distance (TD; m), TD per minute (m), medium-speed running distance (MSR, measurement interval 14.4–19.79 km·h^−1^; m), MSR per minute (m), high-speed running distance (HSR, measurement interval 19.8–25.1 km·h^−1^; m), HSR per minute (m), sprint distance (SPR, >25.2 km·h^−1^; m), SPR per minute (m), acceleration band 1–3 distance (ACC, >1.5 m·s^−2^; m), ACC per minute (m), deceleration band 1–3 distance (DEC, <−1.5 m·s^−2^; m), DEC per minute (m), total micromovements (IMA; n), IMA per minute (n), total player load (PL; AU), and PL per minute (AU). The speed zones were classified according to previous publications [46,47]. In the Catapult system, these speed ranges are default settings for S7 Vector unit. In the case of the player load variable, it is a modified vector that summarizes the change in acceleration in each of the three (X-Y-Z) axes using the following equation [48]:$$\frac{\sqrt{{(a}_{y1}-a_{y-1}{)}^{2}-{(a}_{x1}-a_{x-1}{)}^{2}+{(a}_{z1}-a_{z-1}{)}^{2}}}{100}$$

In line with the suggestion of Maddison and Ni Mhurchu [49], all devices were activated at least 5 min before data collection commenced (before warm-up). All recorded parameters were downloaded from each unit using OpenField Console Software (Catapult Sports, Melbourne, Australia; version 3.9) and subsequently exported as .csv files.

### 2.4. Internal Load Measures

A modified CR-10 Borg scale was employed to measure individual effort and exertion [21]. RPE data were collected 20–30 min after practice [21] via a Facebook Messenger group chat. Players were informed before the research about the 1–10 scale system of RPE, in which “1” represents very weak activity and “10” denotes exceptionally strong activity [50]. This value represents a single number expressed in arbitrary units (AU), later multiplied by the total duration (in minutes) of the session to derive s-RPE [21].

Another measurement used to determine the psychometric status of the players was the Hooper questionnaire, also known as “perceived wellness” [30]. This comprised three items—sleep, stress, and fatigue—with fatigue further divided into central and peripheral components, as suggested by De Dios-Álvarez et al. [34]. The original version included a mood item; however, it was intentionally omitted from this research as although the athletes were asked, in most cases, no reliable information was provided. A five-point Likert scale ranging from 1 to 5 was applied (α = 0.72), where “1” indicated very poor condition and “5” indicated very good condition [51]. It was previously shown to provide an adequate level of internal consistency when applied in soccer [52].

Lastly, we applied the Total Quality of Recovery (TQR) measurement [53] to examine the perceived psycho-physiological recovery of the players, using the same Likert scale ranging from 1 to 5. Perceived wellness indicators and TQR data were collected via a Google Form and sent out through a Facebook Messenger group chat 30 min before training sessions [54]. On match days, only the RPE value was collected to allow players to focus on their primary tasks. All players were previously familiarized with these questionnaires as these were used before (except in the under-15 group). The questionnaire data were collected by the head of the strength and conditioning group at the academy. If a player failed to respond to the questionnaires on the same day it was sent out, those datasets were excluded (even if they responded later).

### 2.5. Statistical Analysis

Data are presented as mean and standard deviation (mean ± SD). The raw data from the .csv files were integrated for statistical analyses using IBM SPSS Statistics software (SPSS v24.0, IBM Corporation, Chicago, IL, USA). To increase statistical efficiency, training sessions and matches were analyzed separately by age groups. The normality of all data was tested using the Shapiro–Wilk test. Both GPS and subjective measures were found to be normally distributed (*p* > 0.05). Cronbach’s alpha was used to test the internal consistency–reliability of the subscale of perceived wellness with an acceptable threshold of (α ≥ 0.70). To verify the correlation between EL and IL, Pearson’s correlation was employed with the alpha level of statistical significance set at *p* < 0.05. The magnitude of the correlation between test measurements was determined according to the following criteria [55]: trivial (r < 0.1), weak (0.1 ≤ r < 0.3), moderate (0.3 ≤ r < 0.5), strong (0.5 ≤ r < 0.7), very strong (0.7 ≤ r < 0.9), and almost perfect (r ≥ 0.9). Additionally, a multiple regression analysis was conducted to predict RPE and s-RPE from GPS-derived variables by examining the coefficients (beta value, R-squared).

## 3. Results

Table 2 shows the descriptive statistics of the GPS-derived variables and RPE, s-RPE, and perceived wellness values.

The correlations between EL and IL for U15, U17, and U19 are illustrated in Figure 1.

Furthermore, perceived wellness values demonstrated a weak correlation with EL-derived parameters across all teams and training sessions.

As shown in Table 3 and Table 4, we distinguished between the results obtained during the training sessions and matches. The under-15 team’s training sessions and matches exhibited a moderate correlation level between RPE and HSR and HSR/min from r = 0.38 to r = 0.34 (*p* < 0.001). Considering both training sessions and matches, only TD showed a strong correlation with s-RPE (r ranging from 0.53 to 0.52, *p* < 0.001). As shown in Figure 2 for U17, RPE demonstrated a very strong correlation with PL (r = 0.71) and s-RPE with TD (r = 0.77), ACC (r = 0.70), DEC (r = 0.76), and PL (r = 0.73) (*p* < 0.001, respectively).

During matches, only TD showed a moderate correlation with s-RPE (r = 0.41). In the U19 team’s training sessions, weak-to-moderate correlations were found for RPE (r ranging from 0.16 to 0.46, *p* < 0.001). Meanwhile, s-RPE showed moderate correlations with TD, DEC, and PL (r = 0.61, *p* < 0.001). During matches, RPE showed the greatest correlation with PL (r = 0.42). Considering s-RPE, significant correlations were observed with TD, MSR, ACC, DEC, and PL (mean r value of 0.59, *p* < 0.001).

For U15, the variable DEC/min (*β* = 2.463, *t*[414] = 2.156, *p* = 0.032) significantly predicted the RPE value. Additionally, SPR (*β* = 1.616, *t*[414] = 2.591, *p* = 0.010), ACC (*β* = 1.376, *t*[414] = 4.140, *p* < 0.001), and DEC (*β* = −3.601, *t*[414] = −3.372, *p* = 0.001) significantly predicted s-RPE. For U17, the analysis indicated that MSR/min (*β* = 0.221, *t*[447] = 3.849, *p* < 0.001) and SPR/min (*β* = 1.186, *t*[447] = 1.990, *p* = 0.047) significantly predicted the RPE score. Furthermore, TD (*β* = 0.170, *t*[447] = 5.533, *p* < 0.001) and MSR/min (*β* = 19.420, *t*[447] = 4.612, *p* < 0.001) significantly predicted the value of s-RPE. Lastly, for U19, SPR/min significantly predicted the RPE value (*β* = 0.48, *t*[515] = 6.247, *p* < 0.001). The best predictors of s-RPE were TD (*β* = 0.146, *t*[515] = 4.873, *p* < 0.001) and SPR/min (*β* = 3.569, *t*[515] = 6.522, *p* < 0.001).

## 4. Discussion

The primary purpose of this study was to identify the relationship between IL values and EL-derived variables and their respective values expressed per minute among U15, U17, and U19 soccer players. Our goal was also to develop a regression model to predict RPE and s-RPE values using GPS variables. The key finding of this study was that there is a difference between the relationship of EL-based variables and IL-based variables between the three age groups. It was also found that, during training sessions, there was a stronger correlation between EL and IL parameters. The findings further prove the importance of TD and s-RPE values for monitoring TL.

Historically, researchers have tended to focus solely on EL-based [9,56] or IL-based [57,58] studies. The main reason for this could be their availability and applicability. Since the introduction of EL-based monitoring advancements in the early 2000s, IL-based studies were the main focus prior to GPS. The importance of the combination of the two types of TL was later proposed by Impellizzeri et al. [59], and these are used together to this day. It is indeed crucial to monitor both EL and IL in soccer [1] to make informed decisions when designing training programs [6].

In recent years, there has been growing interest in investigating the relationship between EL and IL among youth soccer players [4,33,57,60]. Findings from Marynowicz et al. [4] indicate that RPE and s-RPE are significantly related to a large number of EL-based volume and intensity parameters when it comes to calculating one team’s training sessions over an entire season. Specifically, the greatest correlation was found between TD, distance in ACC, and s-RPE (r = 0.70, *p* < 0.001), which is in line with our observations. Despite the small sample size (18 players), this relationship between EL and IL is an important indicator for young footballers aged between 16 and 19. It was also found that there is a higher correlation between s-RPE and EL measurements compared to RPE, which is also supported by our results (see Figure 1), suggesting that s-RPE is a more robust index than RPE. Based on its simple formula, its applicability has been further confirmed, thus simplifying the task of strength and conditioning coaches. Teixeira et al.’s [58] research, which lasted 6 weeks during the first month of the competitive period and involved three age groups (U15, U17, and U19), found small correlations for TD in U15 and U17, while U19 showed moderate correlations for ACC and DEC. However, our findings reveal strong-to-very-strong correlations between TD and s-RPE (U15—r = 0.52, U17—r = 0.74, *p* < 0.001), respectively. The significant difference in results may be attributed to the duration of the studies (6 weeks compared to our 14 weeks). Although it showed that there are some differences in the level of relationship between young age groups with sixty sub-elite players in their study, de Dios-Álvarez et al. [34] found strong-to-very-strong correlation levels (*p* < 0.01) between IL-derived parameters (RPE/s-RPE) and GPS-derived parameters (number of accelerations and decelerations, total distance covered, and player load). They also identified a moderate correlation between TD and RPE when training sessions (r = 0.47) and matches (r = 0.42) were analyzed separately, which is also supported by our findings (see Table 3 and Table 4), noting that TD significantly affects RPE (*p* < 0.001).

Monitoring multiple GPS-derived variables and examining different age groups over an extended period can yield valuable insight that can lead to a better understanding of EL, which has been previously suggested [61] and has been applied in many previous studies since then. Consistent with previous findings, we also observed differences between the three age groups, with the weakest correlation between EL and IL for U15, suggesting that 14–15-year-olds may not yet fully comprehend their own perceptions. Younger players may require a maturational period (late adolescence) to better understand their physical capabilities and bodily perceptions. Additionally, during training sessions, EL-derived volume variables (e.g., TD) show a stronger correlation with s-RPE (U15—r = 0.516; U17—r = 0.768; U19—r = 0.613). Therefore, we concluded that the difference between training and match TL is decisive even for young footballers, which was previously established by a systematic review [62]. Also, as a strength and conditioning coach, it is somewhat monotonous to say that TD is the key parameter that is strongly associated with IL measurements [37], although every head coach tends to be curious about only this, even though countless other interesting variables are available. By the statement of [4], TD can be helpful to monitor TL and consider the transition from youth league to the adult level.

Our regression model indicates that TD, ACC, DEC, MSR/min, SPR/min, and DEC/min can accurately predict IL variables. For both U17 and U19, the SPR/min variable can predict RPE. They explained from 32% to 14% with negligible standard error. In terms of s-RPE prediction, TD appears to be the most valuable source for U15 (R^2^ of 0.55), U17 (R^2^ of 0.73), and U19 (R^2^ of 0.71), which does not align with previous research results [34]. This confirms that the determination of TL is possible without a GPS device by using only RPE values [63].

In this study, the approach to monitor EL and IL showed encouraging results to use these tools on the field. Nonetheless, some limitations must be addressed. One of the main limitations of this research is that although we examined three teams and fifty players, this occurred in a single football academy centre, which limits the generalizability of the results to other populations. Another restriction was that the completion rate of the subjective questionnaires was not perfect, which may affect the overall picture of the players’ TL. Thirdly, although we looked at training sessions separately (contra matches), we did not differentiate between them in terms of the distribution relative to the match. Therefore, future research should examine whether there is a difference in the distribution of different loading days (training). Also, it could be considered to examine the players by playing positions, as there may be differences in the relationship between EL and IL between defenders, midfielders, and strikers. The contextual factors such as opponent, match location, playing home vs. away matches, and match outcome have a significant impact on players’ TL, which may influence the relationship between EL and IL. Additionally, the use of an online application to collect questionnaire responses raises methodological concerns (the athletes did not leave the sport facility at the same time after training sessions, making control more difficult). Thus, it may cause response bias due to time delays, lack of supervision, and/or peer influence among players.

This study aims to provide valuable insights into TL variables and enhance understanding of the correlations between EL and IL in soccer at the elite youth level. Based on our findings, we emphasize the benefits of employing GPS-derived variables in conjunction with self-reported questionnaires (these tools are highly recommended). RPE scores were greatly influenced by contextual factors such as school, weather conditions, and expectations of the player on the field. Hence, not all of these can be controlled by the coach. In this sense, the results suggest that collecting daily RPE and perceived wellness data may not always be necessary as they do not consistently provide a valid and reliable source of information regarding the true IL of the player. One reason for this could be that players do not always report values that correspond with their actual perceptions but rather to fulfil their obligation (see the results for U17 vs. U19). Therefore, to make the measures more adequate and manageable, we recommend less-frequent RPE collection, focusing on key load days such as MD-4, MD+3, MD-3, and MD. Additionally, we suggest assessing perceived wellness on MD+2/MD-5 (typically Monday) to inform practitioners about players’ readiness for training, and on MD-2 (typically Thursday) “Is my athlete ready for further training, or is either a step taper or an intervention needed for recovery?” Finally, we recommend simplifying the perceived wellness questionnaire values by using pictograms (from sad to smiling faces) for young players.

## 5. Conclusions

Consequently, monitoring both EL and IL is essential, providing useful information about TL in relation to training sessions and matches in elite youth soccer. Our findings indicate that perceived wellness measures exhibited weak relationships with EL measures, suggesting that these questionnaire items primarily predict players’ physical and mental readiness, and not IL. The results also demonstrate that both RPE and s-RPE show moderate-to-very-strong correlations with GPS-derived variables, particularly regarding volume metrics.

## Figures and Tables

**Figure 1 sports-13-00246-f001:**
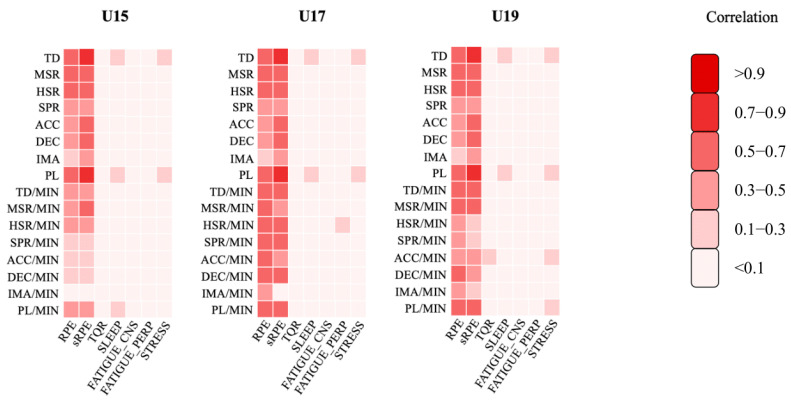
The relationship between IL- and GPS-derived variables among U15, U17, and U19 according to the combination of training sessions and matches.

**Figure 2 sports-13-00246-f002:**
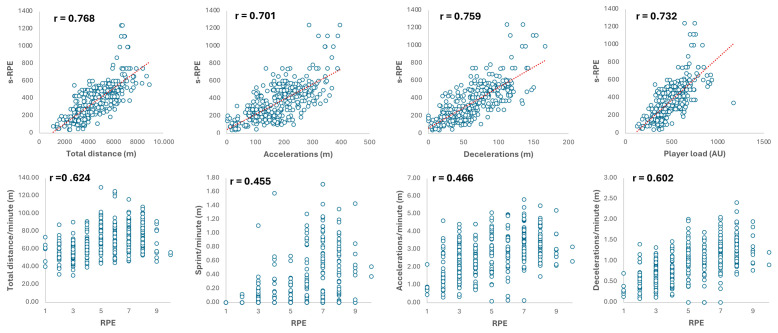
The relationships between s-RPE, RPE, and GPS-derived volume and intensity variables for U17 during training sessions.

**Table 1 sports-13-00246-t001:** Average duration of training sessions and matches including ranges and standard deviations for U15, U17, and U19.

	U15		U17		U19	
	Mean ± SD	Range	Mean ± SD	Range	Mean ± SD	Range
Training	67 ± 12 min	50 to 97 min	65 ± 19 min	52 to 124 min	63 ± 14 min	55 to 102 min
Matches	91 ± 7 min	80 to 98 min	77 ± 6 min	74 to 83 min	91 ± 9 min	81 to 99 min

SD: standard deviation, min = minutes.

**Table 2 sports-13-00246-t002:** Descriptive statistics of collected GPS data.

	U15				U17				U19			
	Training		Matches		Training		Matches		Training		Matches	
	Mean ± SD	Range	Mean ± SD	Range	Mean ± SD	Range	Mean ± SD	Range	Mean ± SD	Range	Mean ± SD	Range
**TD**	4517.64 ± 1186.71	6723.45	8529.77 ± 986.16	4968.98	4413.30 ± 1517.78	7775.48	9766.66 ± 901.89	4113.47	4359.11 ± 1497.82	7163.65	9648.29 ± 1068.61	4751.20
**MSR**	400.21 ± 214.35	1416.99	1295.45 ± 366.36	1817.43	365.57 ± 269.91	1652.98	1421.92 ± 277.11	1147.06	385.17 ± 262.13	2039.74	1372.12 ± 298.46	1340.33
**HSR**	133.01 ± 97.30	660.01	450.67 ± 130.68	562.17	101.47 ± 96.24	469.88	484.24 ± 152.20	579.17	134.20 ± 93.68	426.86	500.24 ± 158.18	703.55
**SPR**	24.22 ± 36.48	247.21	66.18 ± 41.22	163.16	16.74 ± 29.18	141.27	104.75 ± 62.12	258.63	41.52 ± 67.40	414.13	153.15 ± 97.38	365.00
**ACC**	185.73 ± 57.75	273.61	286.96 ± 75.32	313.46	178.56 ± 83.33	393.93	347.66 ± 59.75	257.41	201.76 ± 87.24	557.48	366.47 ± 90.62	439.92
**DEC**	67.66 ± 24.02	136.20	119.55 ± 37.12	145.26	63.35 ± 33.79	166.66	137.34 ± 24.89	101.14	66.36 ± 30.91	159.14	135.56 ± 33.77	148.83
**IMA**	411.31 ± 111.47	796	464.12 ± 173.01	878	416.58 ± 187.09	919	601.82 ± 138.97	618	253.83 ± 101.70	690	384.44 ± 132.08	535
**PL**	504.34 ± 127.39	810.36	927.18 ± 133.95	642.96	477.54 ± 158.58	1045.77	997.07 ± 116.84	536.89	465.66 ± 161.03	859.67	974.41 ± 168.24	729.63
**TD/min**	68.06 ± 13.11	65.80	93.85 ± 6.49	30.71	68.45 ± 16.69	99.30	107.53 ± 9.07	39.63	67.71 ± 16.60	90.06	106.58 ± 9.47	40.28
**MSR/min**	5.96 ± 2.84	16.28	12.17 ± 3.16	14.79	5.63 ± 5.41	45.01	15.70 ± 3.18	11.62	2.06 ± 1.51	7.05	5.58 ± 1.89	8.24
**HSR/min**	1.98 ± 1.37	8.98	4.06 ± 1.17	4.84	1.47 ± 1.51	13.27	5.35 ± 1.73	6.85	0.53 ± 0.94	6.14	1.54 ± 0.96	3.61
**SPR/min**	0.36 ± 0.56	4.01	0.59 ± 0.37	1.57	0.21 ± 0.35	1.71	1.17 ± 0.72	3.37	9.47 ± 22.02	112.42	15.55 ± 21.57	100.59
**ACC/min**	2.79 ± 0.80	4.92	3.16 ± 0.73	3.35	2.72 ± 1.11	5.73	3.83 ± 0.66	2.87	3.09 ± 1.07	6.61	4.06 ± 1.03	4.22
**DEC/min**	1.01 ± 0.31	1.91	1.29 ± 0.34	1.30	0.95 ± 0.44	2.41	1.52 ± 0.30	1.26	1.01 ± 0.40	2.41	1.50 ± 0.40	1.61
**IMA/min**	6.21 ± 1.52	11.14	6.27 ± 2.02	9.59	6.48 ± 2.63	31.12	6.63 ± 1.54	7.75	5.88 ± 2.17	15.97	6.66 ± 2.48	11.49
**PL/min**	7.58 ± 1.34	8.14	10.74 ± 1.19	6.02	7.42 ± 1.84	14.47	10.98 ± 1.32	6.62	7.22 ± 1.80	9.39	10.78 ± 1.81	7.02

SD: standard deviation; TD: total distance; MSR: medium-speed running; HSR: high-speed running; SPR: sprint distance; ACC: acceleration; DEC: deceleration; IMA: inertial movement analysis; PL: player load.

**Table 3 sports-13-00246-t003:** Relationships between s-RPE, RPE, and different GPS-derived variables expressed in volume and per minute regarding training sessions. Value represents correlation coefficients between corresponding variables, respectively.

s-RPE/RPE	U15			U17			U19		
	Value	95% CI	*p*	Value	95% CI	*p*	Value	95% CI	*p*
TD	0.516	0.398–0.598	<0.001	0.768	0.739–0.793	<0.001	0.613	0.552–0.647	<0.001
	0.246	0.106–0.340		0.659	0.605–0.704		0.362	0.255–0.422	
MSR	0.413	0.258–0.491	<0.001	0.537	0.448–0.628	<0.001	0.552	0.487–0.624	<0.001
	0.294	0.144–0.368		0.599	0.550–0.645		0.398	0.328–0.475	
HSR	0.416	0.281–0.500	<0.001	0.641	0.574–0.694	<0.001	0.493	0.425–0.554	<0.001
	0.380	0.276–0.452		0.590	0.529–0.632		0.458	0.374–0.505	
SPR	0.229	0.101–0.343	<0.001	0.621	0.513–0.702	<0.001	0.338	0.169–0.373	<0.001
	0.243	0.245–0.431		0.506	0.421–0.580		0.372	0.305–0.497	
ACC	0.391	0.298–0.461	<0.001	0.701	0.646–0.761	<0.001	0.594	0.536–0.661	<0.001
	0.246	0.142–0.327		0.506	0.547–0.701		0.391	0.342–0.477	
DEC	0.430	0.317–0.525	<0.001	0.759	0.727–0.814	<0.001	0.608	0.548–0.684	<0.001
	0.256	0.138–0.362		0.470	0.640–0.764		0.435	0.370–0.525	
IMA	0.395	0.307–0.472	<0.001	0.513	0.402–0.615	<0.001	0.425	0.365–0.488	<0.001
	0.190	0.083–0.282		0.630	0.350–0.579		0.233	0.136–0.301	
PL	0.496	0.415–0.584	<0.001	0.732	0.668–0.786	<0.001	0.606	0.529–0.645	<0.001
	0.188	0.068–0.270		0.708	0.554–0.690		0.362	0.263–0.419	
TD/min	0.095	–0.089–0.214	0.074 0.040	0.235	0.154–0.358	<0.001	0.320	0.225–0.390	<0.001
	0.153	–0.005–0.270		0.624	0.373–0.533		0.332	0.243–0.408	
MSR/min	0.281	0.056–0.329	<0.001	0.155	0.058–0.366	<0.001	0.305	0.210–0.378	<0.001
	0.252	0.097–0.326		0.369	0.303–0.544		0.421	0.344–0.482	
HSR/min	0.224	0.134–0.377	<0.001	0.288	0.172–0.460	<0.001	0.097	0.012–0.203	0.04 <0.001
	0.345	0.236–0.431		0.435	0.361–0.557		0.261	0.178–0.375	
SPR/min	0.147	0.027–0.252	0.005	0.399	0.323–0.453	<0.001	0.328	0.255–0.433	<0.001
	0.207	0.100–0.302	<0.001	0.455	0.378–0.516		0.454	0.387–0.525	
ACC/min	0.230	−0.071–0.091	0.066	0.325	0.250–0.411	<0.001	0.348	0.263–0.428	<0.001
	0.146	0.021–0.223	<0.001	0.466	0.395–0.552		0.360	0.289–0.447	
DEC/min	0.019	0.002–0.220	0.025	0.458	0.392–0.539	<0.001	0.381	0.291–0.469	<0.001
	0.186	0.071–0.291	<0.001	0.602	0.545–0.668		0.404	0.323–0.497	
IMA/min	−0.050	−0.016–0.054	0.351	0.093	0.025–0.179	0.068	0.153	0.056–0.216	<0.001
	0.064	−0.045–0.150	0.232	0.249	0.153–0.351	<0.001	0.161	0.058–0.230	
PL/min	0.046	−0.091–0.136	0.362	0.147	0.066–0.272	<0.001	0.313	0.203–0.378	<0.001
	0.040	−0.055–0.147	0.165	0.382	0.299–0.474		0.329	0.239–0.397	

SD: standard deviation; TD: total distance; MSR: medium-speed running; HSR: high-speed running; SPR: sprint distance; ACC: acceleration; DEC: deceleration; IMA: inertial movement analysis; PL: player load; RPE: rating of perceived exertion; s-RPE = session-RPE; CI: confidence interval.

**Table 4 sports-13-00246-t004:** Relationships between s-RPE, RPE, and different GPS-derived variables expressed in volume and per minute regarding matches. Value represents correlation coefficients between corresponding variables, respectively.

s-RPE/RPE	U15			U17			U19		
	Value	95% CI	*p*	Value	95% CI	*p*	Value	95% CI	*p*
TD	0.530	0.419–0.661	<0.001	0.415	0.181–0.598	0.001	0.588	0.427–0.698	<0.001
	0.128	−0.074–0.346	0.335	0.119	−0.194–0.382	0.361	0.314	0.115–0.467	0.007
MSR	0.142	0.055–0.393	0.142	0.071	−0.170–0.296	0.588	0.305	0.124–0.428	0.009
	0.007	−0.118–0.262	0.958	0.032	−0.262–0.284	0.809	0.286	0.089–0.447	0.015
HSR	0.026	−0.197–0.270	0.845	0.095	−0.194–0.278	0.466	0.150	−0.133–0.416	0.207
	−0.073	−0.301–0.222	0.585	0.170	−0.179–0.399	0.191	0.293	0.051–0.547	0.012
SPR	0.064	−0.280–0.268	0.629	−0.048	−0.287–0.290	0.716	−0.098	−0.366–0.123	0.415
	0.050	−0.301–0.222	0.626	0.029	−0.202–0.296	0.826	0.003	−0.214–0.234	0.979
ACC	0.061	−0.197–0.338	0.645	0.352	0.060–0.497	0.005	0.275	−0.053–0.475	0.019
	−0.097	−0.261–0.162	0.464	0.320	−0.055–0.502	0.012	0.274	−0.085–0.435	0.020
DEC	−0.052	−0.288–0.157	0.695	0.165	−0.106–0.412	0.205	0.194	−0.128–0.483	0.103
	−0.112	−0.361–0.129	0.399	0.262	−0.040–0.467	0.041	0.239	−0.104–0.488	0.044
IMA	0.049	−0.206–0.385	0.711	0.126	−0.134–0.358	0.334	0.537	0.401–0.649	<0.001
	−0.116	−0.340–0.175	0.380	0.059	−0.283–0.321	0.652	0.433	0.162–0.577	
PL	0.436	0.302–0.595	0.001	0.292	−0.061–0.506	0.022	0.455	0.301–0.627	<0.001
	0.125	−0.096–0.372	0.346	0.084	−0.214–0.342	0.521	0.423	0.275–0.583	
TD/min	0.166	−0.017–0.367	0.209	0.166	−0.583–−0.089	0.209	−0.136	−0.255–0.079	0.255
	0.057	−0.180–0.305	0.671	0.057	−0.453–0.150	0.671	0.059	−0.115–0.285	0.624
MSR/min	0.024	−0.153–0.236	0.858	0.024	−0.499–−0.059	0.858	−0.099	−0.415–0.214	0.408
	−0.037	−0.206–0.247	0.780	−0.037	−0.367–0.177	0.780	0.180	−0.096–0.468	0.130
HSR/min	−0.137	−0.377–0.133	0.302	−0.137	−0.378–0.094	0.302	−0.238	−0.472–0.021	0.045
	−0.108	−0.347–0.210	0.416	−0.108	−0.126–0.334	0.416	−0.049	−0.257–0.182	0.684
SPR/min	−0.005	−0.344–0.220	0.968	−0.005	−0.401–0.204	0.968	−0.020	−0.219–0.225	0.867
	0.067	−0.263–0.274	0.612	0.067	0.256–0.278	0.612	−0.011	−0.272–0.183	0.928
ACC/min	−0.134	−0.358–0.146	0.310	−0.134	−0.308–0.152	0.310	−0.042	−0.422–0.235	0.724
	−0.186	−0.361–0.069	0.158	−0.186	−0.123–0.445	0.158	0.145	−0.250–0.370	0.224
DEC/min	−0.211	−0.474–−0.014	0.109	−0.211	−0.422–−0.013	0.109	0.355	−0.450–0.231	0.002
	−0.161	−0.211–0.109	0.224	−0.161	−0.150–0.339	0.224	0.371	−0.260–0.384	0.001
IMA/min	−0.090	−0.366–0.361	0.500	−0.090	−0.386–0.132	0.500	−0.120	0.187–0.483	0.315
	−0.167	−0.090–0.500	0.205	−0.167	−0.318–0.223	0.205	0.106	0.093–0.504	0.377
PL/min	0.096	−0.109–0.356	0.469	0.096	−0.595–−0.031	0.469	−0.002	−0.174–0.240	0.986
	0.063	−0.187–0.353	0.633	0.063	−0.365–0.107	0.633	0.251	0.093–0.470	0.033

SD: standard deviation; TD: total distance; MSR: medium-speed running; HSR: high-speed running; SPR: sprint distance; ACC: acceleration; DEC: deceleration; IMA: inertial movement analysis; PL: player load; RPE: rating of perceived exertion; s-RPE = session-RPE; CI: confidence interval.

## Data Availability

The data presented are available on request from the corresponding author.

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
