# Peer review of "Relationship Between GPS-Derived Variables and Subjective Questionnaires Among Elite Youth Soccer Players"

_sports, 2025, doi:10.3390/sports13080246_

Round 1
Reviewer 1 Report
Comments and Suggestions for Authors
General comments
This manuscript aimed to investigate the relationships between GPS variables and RPE and quality of recovery questionnaires in youth soccer players. The introduction is well-written, providing relevant background on the manuscript's main topic. The methods are adequately described, needing just some minor adjustments to improve their clarity. The results are well presented and interesting, and they are effectively discussed. Below I have some specific comments.
Specific comments
Methods
How were the speed ranges that classify as medium speed, high speed, and sprinting determined?
Could you give any more details on how playing load is calculated, or what it really means, since it’s given in arbitrary units.
In the last paragraph of the “2.4. Internal load measures” subsection, it states “expect under-15” in parentheses. Should it be “expect,” or is this a typo?
It appears that all statistical analyses you done was separated by age group. Please state that in the statistical analysis subsection.
I’m assuming all variables were normally distributed as you used the Pearson correlation coefficient. If that’s true, please state that.
In the statistical analysis subsection, it’s said that multiple regression analysis was used to predict RPE and s-RPE from the GPS variables. Did you include all GPS variables as predictors in the same model? If so, I would like to see some information regarding the variance inflation factor (VIF) for these variables, as some of them might be collinear.
Also, regarding the regression analysis, it’s affirmed that this analysis was done to predict RPE and s-RPE from the GPS variables. Although a regression model can indeed be used for prediction, from my point of view it appears that your intention was more to understand the relationships between these variables than to actually predict one from another. If this is the case, please change the text to better reflect that.
Results
In the manuscript and in the table abbreviations it says that table 2 have data regarding the RPE and s-RPE. But I couldn’t find these variables in the table.
Are the regression coefficients in tables 3 and 4 the standardized coefficients?
Since data from training sessions and matches were analyzed separately, it could be interesting to add some statistical test to formally assess any difference in the GPS-derived variables between training sessions and matches. That could be included in table 2, for example.
Discussion
In the first paragraph it’s stated that the goal of the regression was to predict the RPE and s-RPE. So, the things I mentioned in my last comment of the methods section also apply here.
Very minor issue, but I think the first and second paragraphs of the discussion should be a single paragraph.
Author Response
Relationship between GPS-derived variables and subjective questionnaires among elite youth soccer players
20 July 2025
We also added a minor change to the perceived wellness, since we did not include mood item (see Materials and methods - Internal load measures - 5th page - 185-187 lines).
Also, we would like to note that we also added the Cronbach alpha reliability test to the perceived wellness questionnaire to demonstrate the consistency of its items (fatigue, sleep, stress). (see Materials and methods - Statistical analysis - 5th page - 188-190 lines) and (see Materials and methods - Statistical analysis - 5th page - 207-208 lines).
Methods
Comment 1: How were the speed ranges that classify as medium speed, high speed, and sprinting determined?
- Response 1: The scientific explanation of speed ranges has been determined by earlier publications (Bradley et al., 2009; Mallo et al., 2015). In the Catapult system, these speed zones (low-medium-high-sprint) are default settings for all devices (S6-S7/8 Vector units) taking into account practical use. Thank you for pointing this out, we added this information to the edited manuscript. (see Materials and methods - External load measures - 4th page - 162-164 lines)
Comment 2: Could you give any more details on how playing load is calculated, or what it really means, since it’s given in arbitrary units.
- Response 2: *Player Load. We've made the requested change in the manuscript. Thank you for reminding us of the importance of clarifying variables that uses arbitrary units. (see Materials and methods - External load measures - 4th page - 164-168 lines)
Comment 3: In the last paragraph of the “2.4. Internal load measures” subsection, it states “expect under-15” in parentheses. Should it be “expect,” or is this a typo?
- Response 3: Typo, that's true, thank you, we've corrected it. (see Materials and methods - Internal load measures - 5th page - 197th line)
Comment 4: It appears that all statistical analyses you done was separated by age group. Please state that in the statistical analysis subsection.
- Response 4: Thank you for the useful comment. We have added it to the statistical analysis subsection. (see Materials and methods - Statistical analysis - 5th page - 204-205 lines)
Comment 5: I’m assuming all variables were normally distributed as you used the Pearson correlation coefficient. If that’s true, please state that.
- Response 5: Indeed, thank you. We had not seen it described in previous research and we thought it would be obvious, when using parametric (Pearson) correlation. Nevertheless, we added it. (see Materials and methods - Statistical analysis - 5th page - 206-207 lines)
Comment 6: In the statistical analysis subsection, it’s said that multiple regression analysis was used to predict RPE and s-RPE from the GPS variables. Did you include all GPS variables as predictors in the same model? If so, I would like to see some information regarding the variance inflation factor (VIF) for these variables, as some of them might be collinear.
- Response 6: Thank you for the comment. Not all variables were used in the same model. For RPE, TD, MSR/min, SPR/min and DEC/min variables were included. For s-RPE, we used TD, ACC, DEC, MSR/min and SPR/min. None of the GPS-based variables exceed a value of 10 in terms of VIF. We can provide these in writing (the exact numbers), if necessary. Additionally, would you suggest us to include these variables in the methods section?
Comment 7: Also, regarding the regression analysis, it’s affirmed that this analysis was done to predict RPE and s-RPE from the GPS variables. Although a regression model can indeed be used for prediction, from my point of view it appears that your intention was more to understand the relationships between these variables than to actually predict one from another. If this is the case, please change the text to better reflect that.
- Response 7: Our secondary goal (with the regression analysis) was indeed to find the variables that can predict the RPE and s-RPE IL variables (subjective). We thought this was necessary from a practical point of view, since most coaches rely only on the TD variable, which is an excellent variable, don't get us wrong, but there are also others that can reflect IL very well, especially for SSGs (small-sided games) and LSGs (large-sided games) tasks in youth soccer. The study did not cover such a wide breakdown (M+/- 5-1), but we think that these variables can also be used with sufficient certainty. The relationship process was our first goal, which was done with correlation analysis.
Results
Comment 8: In the manuscript and in the table abbreviations it says that table 2 have data regarding the RPE and s-RPE. But I couldn’t find these variables in the table.
- Response 8: Indeed, we took that out of the abbreviations. Thank you. (see Results - 6th page - 222nd line)
Comment 9: Are the regression coefficients in tables 3 and 4 the standardized coefficients?
- Response 9: Yes, we used standardized coefficients (beta value) in the study. Should we add to the statistical analyses paragraph or to the tables?
Comment 10: Since data from training sessions and matches were analyzed separately, it could be interesting to add some statistical test to formally assess any difference in the GPS-derived variables between training sessions and matches. That could be included in table 2, for example.
- Response 10: Thank you for the suggestion. We added range to Table 2. We hope this will show better picture on the differences between GPS variables considering training sessions and matches. (see Results - 5-6th pages - between 219th and 220th lines)
Discussion
Comment 11: In the first paragraph it’s stated that the goal of the regression was to predict the RPE and s-RPE. So, the things I mentioned in my last comment of the methods section also apply here.
- Response 11: The same answer as mentioned in the methods above. We think it is important to know which GPS variables can be used to say that perceived effort has changed, the athlete is tired according to their own feeling.
Comment 12: Very minor issue, but I think the first and second paragraphs of the discussion should be a single paragraph.
- Response 12: We edited it. Thank you very much. (see Discussion - 9th page - 273rd line)

Reviewer 2 Report
Comments and Suggestions for Authors
The introduction provides a logical entry into the topic of training loads and the importance of combining EL and IL indicators. Nevertheless, it should be expanded in several key areas:
-
There is a lack of a broader literature review concerning developmental differences in effort perception and training adaptations among young athletes.
-
The practical importance of the EL-IL relationship in coaching practice is insufficiently emphasized.
-
It would be worthwhile to elaborate on the critique of the IL indicators used, such as RPE and s-RPE, and to mention their limitations (e.g., subjectivity, influence of mood, group pressure).
The methodology is largely well-designed and clearly presented. However, there is a lack of detailed information regarding GPS calibration and the analysis of measurement errors. Although the sampling frequencies (10 Hz, 100 Hz) are provided, the margin of error and any methods for compensating GPS drift should also be included. Furthermore, the method used for collecting RPE (via Messenger) raises certain methodological concerns. The authors should discuss the potential distortion of responses due to time delays, lack of supervision, or peer influence among players. Additionally, the approach to handling missing values is not clearly defined - for example, in cases where a player did not respond to the RPE or failed to complete the wellness questionnaire.
The authors mentioned certain limitations, but this list should be expanded. It may be worth considering, for instance:
-
The limited number of teams and a single academy center, which restricts the generalizability of the findings to other populations of youth football players;
-
The lack of comparative analysis between playing positions - different positions generate different EL profiles, which may affect their relationship with IL;
-
The absence of an examination of the impact of the opponent and match context (e.g., result, difficulty of the opponent, playing home vs. away) on the load.
Author Response
Relationship between GPS-derived variables and subjective questionnaires among elite youth soccer players
20 July 2025
We would like to thank the Reviewer for the time and energy he/she devoted to this manuscript. We provide our responses to the comments and suggestions below.
We also added a minor change to the perceived wellness, since we did not include mood item (see Materials and methods - Internal load measures - 5th page - 185-187 lines).
Also, we would like to note that we also added the Cronbach alpha reliability test to the perceived wellness questionnaire to demonstrate the consistency of its items (fatigue, sleep, stress). (see Materials and methods - Statistical analysis - 5th page - 188-190 lines) and (see Materials and methods - Statistical analysis - 5th page - 207-208 lines).
The introduction provides a logical entry into the topic of training loads and the importance of combining EL and IL indicators. Nevertheless, it should be expanded in several key areas:
- There is a lack of a broader literature review concerning developmental differences in effort perception and training adaptations among young athletes.
- Response 1: Thank you very much! We absolutely agree with the first part, and we added it to the introduction section, supported by literature. However, we have some concerns about the second part, because we did not look for adaptation process in this study, it meant to be investigated training load and its relation to each other (EL-IL). If there is any objection to this, we are happy to accept it and analyze it in more depth across literature. (see Introduction - 2nd page - 57-61 lines)
- The practical importance of the EL-IL relationship in coaching practice is insufficiently emphasized.
- Response 2: Thank you very much! We included it in the introductory section with one reference. (see Introduction - 2nd page - 70-72 lines)
- It would be worthwhile to elaborate on the critique of the IL indicators used, such as RPE and s-RPE, and to mention their limitations (e.g., subjectivity, influence of mood, group pressure).
- Response 3: Thank you very much! We have highlighted these with references. (see Introduction - 2nd page - 63-66 lines)
The methodology is largely well-designed and clearly presented. However, there is a lack of detailed information regarding GPS calibration and the analysis of measurement errors. Although the sampling frequencies (10 Hz, 100 Hz) are provided, the margin of error and any methods for compensating GPS drift should also be included. Furthermore, the method used for collecting RPE (via Messenger) raises certain methodological concerns. The authors should discuss the potential distortion of responses due to time delays, lack of supervision, or peer influence among players. Additionally, the approach to handling missing values is not clearly defined - for example, in cases where a player did not respond to the RPE or failed to complete the wellness questionnaire.
- Response 4: Thank you very much! We added a few variables (HDOP, average GNSS, number of satellites) to the research that relate to the strength of GPS signal. If the signal was weak (bad weather conditions, coverage, signal jammers) then we excluded those datasets. We hope this gives a more accurate picture of its accuracy, referring to a literature in the manuscript (see Materials and methods - External load measures - 4th page - 146-151 lines). Indeed, questionnaires sent via Messenger are problematic. In previous research they used the WhatsApp application (de Dios-Álvarez et al., 2023), which we know it's not an excuse. This has more practical utility, and with proper instructions, the players knew their task. The results of the U19 team refute this, we know, so it is not a gold standard. We are still thinking about what would be the best. Maybe it is worth considering the local conditions. We added - if someone failed to answer, how we managed it (see Materials and methods - Internal load measures - 5th page - 198-200 lines), and also to the discussion part (see Discussion - 10th page - 348-352 lines).
The authors mentioned certain limitations, but this list should be expanded. It may be worth considering, for instance:
- The limited number of teams and a single academy center, which restricts the generalizability of the findings to other populations of youth football players;
- Response 5: Thank you for your suggestion. Absolutely agree on this, we have added it to the discussion part. Just a note: we think the number of teams and players are quite acceptable, as most academies work with about three or four teams, with an average of twenty players. (see Discussion - 10th page - 336-339 lines).
- The lack of comparative analysis between playing positions - different positions generate different EL profiles, which may affect their relationship with IL;
- Response 6: Thank you! We agree on this when it comes to training sessions. We have added it to the discussion section (see Discussion - 10th page - 344-346 lines). Since we examined academy soccer players, we did not differentiate by player position (purposefully), as although the main playing positions does not change rapidly, whether someone is a defender, midfielder or attacker, but within these there can be changes during matches (e.g. from defensive midfielder to winger due to few substitution).
- The absence of an examination of the impact of the opponent and match context (e.g., result, difficulty of the opponent, playing home vs. away) on the load.
- Response 7: Thank you very much! We completely agree on this one as well, we added it to the discussion section (see Discussion - 10th page - 346-348 lines).

Round 2
Reviewer 2 Report
Comments and Suggestions for Authors
The authors have addressed all the suggestions outlined in my review, making appropriate revisions to both the theoretical framework and interpretative sections. In its current form, the article meets the journal's academic standards and is suitable for publication.